# Anti-Hyperalgesic Efficacy of Acetyl L-Carnitine (ALCAR) Against Visceral Pain Induced by Colitis: Involvement of Glia in the Enteric and Central Nervous System

**DOI:** 10.3390/ijms241914841

**Published:** 2023-10-02

**Authors:** Elena Lucarini, Laura Micheli, Alessandra Toti, Clara Ciampi, Francesco Margiotta, Lorenzo Di Cesare Mannelli, Carla Ghelardini

**Affiliations:** Department of Neuroscience, Psychology, Drug Research and Child Health (NEUROFARBA), Pharmacology and Toxicology Section, University of Florence, 50139 Florence, Italy; laura.micheli@unifi.it (L.M.); alessandra.toti@unifi.it (A.T.); clara.ciampi@unifi.it (C.C.); francesco.margiotta@unifi.it (F.M.); lorenzo.mannelli@unifi.it (L.D.C.M.); carla.ghelardini@unifi.it (C.G.)

**Keywords:** acetyl L-carnitine, colitis, visceral pain, enteric neuron, enteric glia, astrocyte

## Abstract

The management of abdominal pain in patients affected by inflammatory bowel diseases (IBDs) still represents a problem because of the lack of effective treatments. Acetyl L-carnitine (ALCAR) has proved useful in the treatment of different types of chronic pain with excellent tolerability. The present work aimed at evaluating the anti-hyperalgesic efficacy of ALCAR in a model of persistent visceral pain associated with colitis induced by 2,4-dinitrobenzene sulfonic acid (DNBS) injection. Two different protocols were applied. In the preventive protocol, ALCAR was administered daily starting 14 days to 24 h before the delivery of DNBS. In the interventive protocol, ALCAR was daily administered starting the same day of DNBS injection, and the treatment was continued for 14 days. In both cases, ALCAR significantly reduced the establishment of visceral hyperalgesia in DNBS-treated animals, though the interventive protocol showed a greater efficacy than the preventive one. The interventive protocol partially reduced colon damage in rats, counteracting enteric glia and spinal astrocyte activation resulting from colitis, as analyzed by immunofluorescence. On the other hand, the preventive protocol effectively protected enteric neurons from the inflammatory insult. These findings suggest the putative usefulness of ALCAR as a food supplement for patients suffering from IBDs.

## 1. Introduction

Chronic abdominal pain is a cross-cutting symptom of gastrointestinal disease. It frequently develops in patients who have experienced inflammatory intestinal damage in both childhood and adulthood. Accordingly, about 20–40% of patients with inflammatory bowel disease (IBD) continue to complain of abdominal pain and intestinal dysfunction even after reaching clinical remission [1]. Unfortunately, few therapeutic interventions are available for the treatment of this kind of abdominal pain, and most of them are characterized by limited efficacy and safety. Indeed, the anti-inflammatory and immunomodulatory treatments used in IBD patients are often not as effective in preventing the development of visceral hypersensitivity as they are in controlling intestinal pathology [2]. The difficulty in the identification of effective medicines is closely related to the multiplicity of pathological aspects related to post-inflammatory persistent pain consisting of nociceptive, inflammatory, and neuropathic components as well as the involvement of the immune system [3]. 

Acetyl-L-carnitine (ALCAR) is configured as a potentially suitable tool since its beneficial effects intercept many of the systems involved in the pathophysiology of this kind of pain. In addition to its well-defined role in energy metabolism, ALCAR has antioxidant, neuromodulator, neuroprotective, and regulatory properties of gene expression, together with an excellent safety and tolerability profile [4,5,6,7]. 

The scientific literature reports carnitine derivatives as efficient protectors against ulcerative colitis and in some models of experimental colitis [8,9]. Moreover, the presence of the acetyl group in the ALCAR molecule can enhance cholinergic signalling by promoting the synthesis of the neurotransmitter acetylcholine, which plays an important role in both the enteric and central nervous systems [10,11,12]. Interestingly, it has been observed that acetylcholine signalling has significant antinociceptive effects in the development of visceral pain [13,14,15], so it has been proposed as a therapeutic target [16]. Cholinergic signalling is mainly attributable to the activation of the vagus nerve, which mediates a dynamic interaction between the brain and the intestine capable of containing the inflammatory response through humoral and neural signals and capable of performing an important control function even in chronic pathologies affecting the gut [17]. After integration in the central autonomic network of peripheral sensations, an efferent response through modulation of preganglionic parasympathetic neurons of the dorsal motor nucleus of the vagus and/or preganglionic sympathetic neurons of the spinal cord is able to modulate gastrointestinal nociception, motility, and inflammation [18]. ALCAR might potentially modulate visceral pain by interfering with both the intestinal and spinal cholinergic signalling pathways involved in the regulation of visceral sensitivity. This indication fits perfectly with ALCAR, which in animal models of chronic (neuropathic) pain has shown analgesic and neuroprotective efficacy dependent precisely on the activation of nicotinic receptors [19]. 

The aforementioned theoretical bases for the use of ALCAR in the treatment of persistent visceral pain accompany the multiple evidence of the efficacy of ALCAR against acute nociceptive pain as well as persistent pain of various aetiologies [11,19,20,21,22,23,24,25]. Many of these studies have also demonstrated versatile protective capacity against tissue alterations, which are the basis of chronic pain. Intriguingly, ALCAR seems to have a particular trophism for the metabolism of glial cells [26,27,28,29], which are involved in the mechanisms of sensitization at the base of pain chronicity at both the peripheral and central levels [3,30,31]. 

The aim of the present research was to evaluate the anti-hyperalgesic efficacy of the repeated administration of ALCAR in a model of persistent visceral pain resulting from colitis induced by the intrarectal injection of 2,4-dinitrobenzene sulfonic acid (DNBS) in rats. In the same animals, ALCAR neuroprotective effects on both peripheral and central nervous system were investigated. 

## 2. Results

### 2.1. Anti-Hyperalgesic Efficacy of ALCAR on Visceral Pain Induced by DNBS in Rats

Figure 1 shows the effect of the repeated administration of ALCAR on visceral hypersensitivity induced by 2,4-dinitrobenzene sulfonic acid (DNBS) intrarectal injection in rats. Visceral sensitivity was assessed by measuring the Abdominal Withdrawal Reflex (AWR, Figure 1C,D) and the Viscero-Motor Response (VMR, Figure 1E,F) to colorectal distension (0.5–3 mL) in the acute (Day 7) and in the post-inflammatory phase of colitis (Day 14) caused by DNBS. As reported in the experimental scheme (Figure 1A), two different protocols (preventive and interventive) were adopted to evaluate the effect of the repeated administration of ALCAR on both the pain development and persistence. In the preventive protocol (green line), rats were daily administered with ALCAR starting 14 days to 24 h before the delivery of DNBS. In the interventive protocol (blue line), instead, the rats were daily administered with ALCAR starting the day of DNBS injection and continuing the treatment for 14 days. The repeated administration of ALCAR did not exert significant effects on the animal’s body weight; indeed, the loss of weight observed in DNBS animals treated with ALCAR was comparable to that of DNBS animals receiving the vehicle. Yet, ALCAR treatment did not affect the body weight of healthy animals once administered before DNBS injection (Figure 1B).

The injection of DNBS caused a significant increase in both the AWR and the VMR responses to colorectal distension, which was evident in the acute phase (day 7) and persisted after colitis resolution (day 14). The preventive treatment with ALCAR attenuated visceral hypersensitivity establishment after colitis induction. On both day 7 and day 14, the AWR score of the DNBS + ALCAR group was significantly lower than that of the DNBS + vehicle group (Figure 1C and Figure 1D, respectively). A lower VMR response to colorectal distension was also observed in the animals treated with ALCAR (Figure 1E,F), though this effect was significant only on day 14 (Figure 1F). The interventive treatment with ALCAR showed greater efficacy than the preventive protocol in counteracting visceral pain development and persistence after DNBS injection. Indeed, in the acute phase of colitis (day 7), both the AWR and the VMR to colorectal distension resulted in significantly lower in the DNBS animals receiving ALCAR with respect to those receiving the vehicle (Figure 1C and Figure 1E, respectively). As a result of the interventive approach, DNBS + ALCAR animals, also displayed significantly reduced post-inflammatory visceral hypersensitivity, as emerged by both the AWR and VMR tests (Day 14, Figure 1D,F). 

### 2.2. Effect of ALCAR on Colon Damage and Mast Cell/Eosinophils Infiltration Caused by DNBS Injection

On day 14, after the behavioural test, the animals were sacrificed to evaluate colon damage. A macroscopic damage score was assigned to the fresh tissue based on the stool consistency, the presence of hyperaemia or ulcers, and the thickness of the colonic walls. Figure 2 shows the effect of the ALCAR-based preventive (green column) and interventive (blue column) treatments on the intestinal damage induced by DNBS injection in rats (red column). Both the preventive and interventive treatments were able to significantly improve colon damage from a macroscopic point of view (Figure 2A), but only the interventive protocol effectively reduced the microscopic alterations induced by DNBS (Figure 2B). Regarding the reduction in colon length caused by colitis, no differences were detected in DNBS animals receiving the ALCAR with respect to the DNBS + vehicle group (Figure 2C). Looking at H&E-stained colon slices (Figure 2D and Appendix A), it was observed that after the interventive treatment with ALCAR, the submucosa of DNBS animals was significantly less thick in comparison to that of DNBS + vehicle animals (black double arrows), the mucosa was almost restored, and the inflammatory infiltrate was restricted to the submucosa (white arrows). An increased mast cell and eosinophil infiltration within the colon is one of the peripheral mechanisms that has been associated with visceral sensitivity persistence [3,32], and it is usually reduced by treatment that prevents pain establishment by protecting the colon from the inflammatory insult [30,33,34]. Regarding the effect of ALCAR, neither the preventive nor the interventive protocol had any effect on mast cell and eosinophil infiltration in the submucosae of DNBS-treated animals (Figure 2E,F), as clearly emerges in the representative picture, showing mast cell granules stained in purple and eosinophils stained in pink with GIEMSA (Figure 2G and Appendix A).

### 2.3. Neuroprotective Effects of ALCAR on the Enteric Nervous System

The damage to enteric neurons, as well as the remodelling of enteric neuronal networks, contributes to the development of post-inflammatory visceral hypersensitivity in animals [35,36,37]. The impairment of the enteric nervous system caused by intestinal inflammation is attested to the slight but significant reduction in the immunoreactivity of the neuronal marker PGP 9.5 observed in the colon myenteric plexus of DNBS-treated animals (Figure 3A), though the number of PGP9.5-positive cells was not affected (Figure 3B). The preventive treatment with ALCAR protected enteric neurons from the inflammatory insult. Indeed, PGP9.5 immunoreactivity in the myenteric plexus of these animals was significantly augmented. By contrast, the interventive treatment was ineffective in neuronal damage (Figure 3A, illustrative images in Figure 3E and Appendix A).

Additionally, enteric glia surrounding neurons in the gut can undergo extensive changes after an inflammatory insult, which in turn alters the interaction with the heterogenous cellular environment, contributing to visceral hyperalgesia [30,38]. Within the colon, an increased immune reactivity of GFAP, reflecting enteric gliosis, is detectable in DNBS-treated rats (Figure 3C). The number of enteric glial cells was not affected by any treatment (Figure 3D). The interventive treatment with ALCAR significantly attenuated the activation of enteric glia caused by DNBS-induced colitis. On the other hand, the preventive treatment was associated with a drastically enhanced expression of GFAP in the myenteric plexus (3C, illustrative images in Figure 3E and Appendix A).

### 2.4. Neuroprotective Effects of ALCAR on Central Nervous System

Fourteen days after DNBS injection, immunofluorescence analysis performed on lumbosacral sections of the spinal cord revealed the presence of gliosis (Figure 4A,B,D,E), a phenomenon involved in central sensitization that accompanies several chronic painful conditions [3,39]. The dorsal horns of DNBS-treated animals were characterized by astrocyte and microglia activation (GFAP and Iba-1-positive cells, respectively). Both GFAP immunoreactivity (Figure 4A) and the density of astrocytes (GFAP-positive cells; about 40%; Figure 4B) were found to be significantly increased in the dorsal horns of DNBS-treated animals. The interventive treatment with ALCAR significantly reduced GFAP-related immunoreactivity (Figure 4A) and astrocyte density (Figure 4B), while the animals receiving the preventive protocol were comparable to the DNBS + vehicle group (Figure 4A,B). Illustrative images of each experimental group were reported in Figure 4C and Appendix A.

The immunoreactivity related to Iba-1 was 2.5-fold higher in DNBS + vehicle-treated animals as compared to controls (Figure 4D): microglia (Iba1-positive cells) did not change in density (Figure 4E) but underwent well-defined morphological alterations (illustrative images in Figure 4F and Appendix A). In particular, the activated status of microglial cells in the DNBS + vehicle group was recognized by the loss of the processes that are peculiar to resting conditions, the enlargement of the body size, and the increase in the expression of the marker [40]. Both the preventive and the interventive treatment with ALCAR did not influence Iba-1 immunoreactivity in DNBS-treated animals (Figure 4D). Anyway, in DNBS animals receiving ALCAR, unlike those receiving the vehicle, the increase in Iba-1 immunoreactivity was associated with an increased density of microglia within the dorsal horns of the spinal cord (Figure 4E), rather than with morphological changes (DNBS illustrative images in Figure 4F and Appendix A). 

## 3. Discussion

The present work highlights for the first time the potential of ALCAR subcutaneous administration as a therapeutic intervention against colitis-induced persistent visceral pain. Moreover, the histological analysis performed on the both enteric and central nervous system attested the capacity of ALCAR to modulate glial cell activity, a mechanism that might partially explain its anti-hyperalgesic efficacy. Indeed, a peculiarity of ALCAR is that the effects on pain were independent of tissue healing since only a slight attenuation of inflammatory damage to the colon was detected after the treatment.

To date, there is no direct evidence in the literature attesting the ALCAR efficacy on pain of gastrointestinal origin. ALCAR was reported to alleviate mechanical and heat hypersensitivity in a mouse model of pancreatitis characterized by microglial activation along the brain’s pain circuitry. Though visceral hypersensitivity was not directly investigated in that study, an effect similar to that observed in the model of colitis might occur [41]. As mentioned in the introduction, beyond its primary role in energy metabolism, ALCAR is endowed with antioxidant properties, protects from oxidative stress, modulates several neurotransmitters, such as acetylcholine, serotonin, and dopamine, and neurotrophic factors, such as nerve growth factor and metabotropic glutamate (mGlu) receptors, by means of epigenetic mechanisms [42]. In vivo, ALCAR supplementation displayed long-term neurotrophic and analgesic activity in a variety of experimental models of chronic inflammatory and neuropathic pain. By inducing the expression of mGlu2 receptors (mGlu2Rs) at nerve terminals, ALCAR might give rise to visceral analgesia and also prevent spinal sensitization, a mechanism already documented in the literature [43]. In particular, the up-regulation of mGlu2R mediated by ALCAR takes a few days to become established but lasts for several weeks after treatment interruption [44,45]. In a recent study, ALCAR was found to induce analgesia in mice modelling Fabry disease [46], an X-linked lysosomal storage disorder caused by deficient function of the alpha-galactosidase A (α-GalA) enzyme which leads to multisystemic clinical manifestations, among them neuropathic pain and gastrointestinal dysfunctions that appears in the early stage of the disease [47,48]. The authors demonstrated that the ALCAR analgesic effect was mediated by an up-regulation of mGlu2Rs in cultured DRG neurons isolated from 30-day ALCAR-treated α-GalA KO mice. Anyway, since the up-regulation of mGlu2 receptors was no longer present in DRG neurons isolated 30 days after the end of treatment, despite the persistence of analgesia, the authors concluded that ALCAR long-lasting analgesia was maintained by additional mechanisms [46]. Indeed, several mechanisms are probably involved in the effects of ALCAR on pain, one among all the regulation of M1 muscarinic receptor activity in the central cholinergic system [11,21]. Noteworthy, the intracellular events triggered by the activation of the muscarinic system are blocked also by the Glu2 receptor antagonism [49], suggesting a correlation between the glutamatergic and muscarinic systems, with the muscarinic system acting downstream of the mGlu2 receptors upregulation. Similar long-term mechanisms might explain the anti-hyperalgesic effect of ALCAR-based preventive intervention observed in our experimental model. 

From our findings, it clearly emerges that the subcutaneous administration of ALCAR has no significant anti-inflammatory effect on the colon, while systemic neuroprotection is conserved, as attested by both colon and spinal cord immunofluorescence analysis. The mechanisms behind these neuroprotective effects were not deeply investigated, but some hypotheses can be advanced based on the extensive literature on ALCAR. L-carnitine and ALCAR are both antioxidants, energy suppliers, and neuroprotective agents [42,50,51]. ALCAR mimics the anticonvulsive effects of fasting and the ketogenic diet by serving as an alternative energy source for brain cells with altered amino acid homeostasis [52]. The increased plasma levels of ALCAR in children on the ketogenic diet [53], suggested a possible involvement of ALCAR in the stabilization of neuronal activity, which might prevent nervous tissue from the phenomenon of excitotoxic neuronal damage involved in chronic pain transition [54,55,56]. A protective effect of ALCAR on the apoptotic pathway of peripheral neuropathy was described in a rat model of peripheral neuropathy obtained by the loose ligation of the rat sciatic nerve [22]. The same authors reported that the anti-neuropathic effect of ALCAR was prevented by cotreatment with the nicotinic antagonist mecamylamine, broadening the spectrum of action of ALCAR to the nicotinic system [19]. Interestingly, Costa et al. showed that both acute and chronic oral treatments with nicotine remarkably inhibited mechanical allodynia in the mouse model of colitis induced by dextran sulfate sodium (DSS) administration. Nevertheless, nicotine did not affect DSS-induced colonic damage and inflammation [57]. Similarly, ALCAR counteracted visceral pain caused by DNBS colitis with a poor effect on the inflammatory damage. The antinociceptive effect of nicotine in DSS mice was dependent on the selective activation of α7 nAchR [57]. Di Cesare Mannelli et al. (2014) demonstrated that repeated treatment with selective alpha7 nAChR agonists can protect nervous tissues (dorsal root ganglia and peripheral nerves) from oxaliplatin-induced damage. Moreover, the authors reported that alpha7 nAChR stimulation increased glial cell number (astrocytes and microglia) in a region-specific manner either in naïve and neuropathic rats, an effect correlating with the anti-hyperalgesic efficacy of the alpha7 nAChR agonists [58]. Here, we observed that ALCAR treatment increased the density of microglial cells in the spinal cord, counteracting the overactivation of astrocytes at the same time. Previous studies conducted in vitro suggested that the treatment of activated microglia with l-carnitine could reverse the effects of detrimental neuroinflammation [59]. Despite the apparent contrast, both these results attest to a role for ALCAR in the modulation of microglia phenotype which might assume either a positive or a negative meaning according to the context [60,61,62,63]. This evidence contributes to supporting the thesis that fine regulation of neuro-immune cross-talk is responsible for ALCAR efficacy against chronic pain. 

It is also important to consider that the effects of ALCAR might be mediated by itself or by the metabolism to L-carnitine and acetyl. Carnitine is essential for the transfer of long-chain fatty acids across the inner mitochondrial membrane for subsequent β-oxidation [64], which preferentially takes place in astrocytes in the nervous tissues [65,66]. Yet, ALCAR, providing acetyl-CoA for energy metabolism, might potentiate astrocytic fatty acid metabolism and prevent astrogliosis induced by stress conditions, like those resulting from a continuous stimulation of neurons involved in pain signalling. In support of this, the treatment with N-acetyl-cysteine and ALCAR was found to attenuate neuroinflammation, induce axonal sprouting, and reduce the death of motoneurons in a preclinical model of spinal cord injury [67]. This might explain the neuroprotective and anti-hyperalgesic efficacy of the ALCAR-based interventive protocol, which was associated with a significant reduction of enteric gliosis in the colon myenteric plexus and of reactive astrogliosis at the spinal cord level, as attested by the analysis of GFAP immunoreactivity and by the number of GFAP-positive cells. Enteric glia in the enteric nervous system, as astrocytes in the central nervous system, surround neurons, modulating their activity but also mediating the interaction between neurons and immune cells, assuming a key role in visceral pain [3,30,38]. Within the gut, the enteric glia responds to the perturbations of the environment to protect the nervous system from damage and maintain local homeostasis. However, during inflammatory conditions, enteric glial activity undergoes extensive changes that alter its interaction with surrounding cells and contribute to the persistence of visceral hypersensitivity in different ways [68,69,70]. In the central nervous system, astrocytes undertake critical roles in the pathophysiology of chronic pain. Through the secretion of transmitters, reactive astrocytes can influence primary afferent neuronal signalling or sensitize second-order neurons in the spinal cord. In addition, astrocytes can alter pain perception through the nociceptive pathway by creating astrocytic networks capable of transducing signals for extended distances across and along the spinal cord, up to the brain [30,71]. Noteworthy, spinal reactive astrogliosis has been associated with persistent visceral pain resulting from colitis in rats [3]. 

On the other hand, with the ALCAR-based preventive protocol, a long-lasting anti-hyperalgesic efficacy has been achieved, and it was associated with a direct neuroprotective effect on enteric neurons, as attested by the protection from the neuronal damage caused by colitis and the further increase in PGP9.5 expression (neuronal marker) in the myenteric plexus. On the other hand, the treatment with ALCAR might be unable to elicit an increase in the expression of PGP9.5 by neurons under the inflammatory state (as that caused by DNBS injection), when the cell apparatus is dedicated to other functions. Indeed, there is no difference in the expression of PGP9.5 between the group of DNBS animals receiving the vehicle and those receiving the ALCAR (interventive). In the case of the preventive intervention, it is important to take into account that the protective effects of ALCAR against pain might take place in a phase in which the animals have not yet to the inflammatory insult (physiological condition). Indeed, it is likely that the increased PGP9.5 expression elicited by ALCAR treatment takes place before the injection of DNBS and contributes to protect neurons from the inflammatory insult. It might be alternatively possible that the pre-treatment with ALCAR modified the neuronal gene expression patterns, predisposing neurons to a different functional output during inflammation. At the same time, ALCAR may need time to exert certain beneficial effects. Even if further investigations are needed to highlight the mechanisms, the protection triggered by the preventive administration of ALCAR seems to cover a long time after the interruption of the treatment, providing a shield against the deleterious consequences of inflammation. Although the pleiotropic activity of ALCAR represents a strength in the therapy of visceral pain, it makes it difficult to study single mechanisms in specific contexts since the effect of repeated administration with ALCAR might be the result of concomitant changes. The investigation of the mechanisms requires dedicated functional studies, which, based on the evidence collected in this work, might feed into future research.

In conclusion, despite its limitations, the presented work highlighted another therapeutic application for ALCAR, which might be integrated into the therapy of IBDs as an adjuvant strategy to either prevent or counteract abdominal pain establishment. 

## 4. Materials and Methods

### 4.1. Animals

Male Sprague-Dawley rats (Envigo, Varese, Italy), weighing approximately 250 g at the beginning of the experimental procedure, were used. Animals were housed in CeSAL (Centro Stabulazione Animali da Laboratorio, University of Florence, Florence, Italy) and used at least 1 week after their arrival. Four rats were housed per cage (size 26 × 41 cm); animals were fed a standard laboratory diet and tap water ad libitum and kept at 23 ± 1 °C with a 12 h light/dark cycle, light at 7 a.m. All animal manipulations were carried out according to the European Community guidelines for animal care (DL 116/92) and the application of the European Communities Council Directive of 24 November 1986 (86/609/EEC). The ethical policy of the University of Florence complies with the Guide for the Care and Use of Laboratory Animals of the US National Institutes of Health (NIH Publication No. 85-23, revised 1996; University of Florence assurance number: A5278-01). Formal approval to conduct the experiments described was obtained from the Animal Subjects Review Board of the University of Florence. Experiments involving animals have been reported according to ARRIVE guidelines [72]. All efforts were made to minimize animal suffering and to reduce the number of animals used.

### 4.2. Experimental Design 

Colitis was induced in accordance with the method described previously [73]. In brief, during short anaesthesia with isoflurane (2%), 15 mg of 2,4-dinitrobenzene sulfonic acid (DNBS; Sigma-Aldrich, Milan, Italy) in 0.25 mL of 50% ethanol was administered intrarectally via a polyethylene PE-60 catheter inserted 8 cm proximal to the anus. Control rats received 0.25 mL of vehicle. A single injection of DNBS or vehicle was performed on day 0 as represented in Figure 1A. ALCAR (100 mg kg^−1^) was dissolved in saline solution and twice daily administered by subcutaneous injections. Two different protocols were adopted to evaluate the effect of the repeated administration of ALCAR. In the preventive protocol, rats were daily administered with test drugs 14 days to 24 h before DNBS. In the intervention protocol, daily drug treatment started on the day of DNBS delivery and continued until rats were sacrificed on day 14 (as described in the experimental scheme, Figure 1A). The effect of the preventive and interventive protocols was separately examined in order to deepen the long-term protective efficacy of ALCAR against the development of pain, which might further support its employment as a maintenance therapy in patients. This strategy was chosen to evaluate if ALCAR had beneficial effects independent of the regulation of the inflammatory process in order to develop a selective therapy for pain.

### 4.3. Assessment of Visceral Sensitivity by Visceromotor Response (VMR)

For the electromyographic (EMG) recordings, in animals under anaesthesia (isoflurane 2%) two electrodes (AS631, Cooner Wire, Chatsworth, CA, USA) were sewn into the external oblique abdominal muscle and exteriorized dorsally. The visceromotor response (VMR) to Colo-Rectal Distension (CRD) was recorded in the animals under anaesthesia (isoflurane 2%) as previously described [3]. To perform colorectal distension, a balloon (length: 4.5 cm) was inserted into the colon and filled with increasing volumes of water (0.5, 1, 2, 3 mL); 5 min was the time elapsed between two consecutive distensions. This paradigm was used for the evaluation of visceral sensitivity over time and the assessment of drug efficacy. A smaller balloon (length: 2 cm) was used to perform rectal distension by its positioning in the rectum and filling it with 1.5 mL (this paradigm was applied to evaluate visceral sensitivity in a distal region of the gut rather than that affected directly by DNBS damage). During the test, the electrodes were connected to a data acquisition system, and the corresponding EMG signals, consequent to colorectal stimulations were recorded, amplified, and filtered (Animal Bio Amp, ADInstruments, Colorado Springs, CO, USA), digitized (PowerLab 4/35, ADIinstruments), analysed, and quantified using LabChart 8 (ADInstruments). To quantify the magnitude of the visceromotor response at each distension volume, the area under the curve (AUC) immediately preceding the distension (30 s) was subtracted from the AUC during the balloon distension (30 s), and the responses were expressed as per cent increments from the baseline. 

### 4.4. Assessment of Visceral Sensitivity by Abdominal Withdrawal Reflex (AWR)

Visceral sensitivity to colorectal distension (CRD) was assessed via Abdominal Withdrawal Reflex (AWR) measurement using a semi-quantitative score as described previously in conscious animals [3]. Briefly, rats were anaesthetized with isoflurane, and a lubricated latex balloon (length: 4.5 cm), attached to polyethylene tubing, assembled to an embolectomy catheter, and connected to a syringe filled with water, was inserted through the anus into the rectum and descending colon of adult rats. The tubing was taped to the tail to hold the balloon in place. Then rats were allowed to recover from the anaesthesia for 15 min. AWR measurement consisted of visual observation of animal responses to graded CRD (0.5, 1, 2, 3 mL) by a blinded observer who assigned scores: No behavioural response to colorectal distention (0); Immobile during colorectal distention and occasional head clinching at stimulus onset (1); Mild contraction of the abdominal muscles but the absence of abdomen lifting from the platform (2); observed strong contraction of the abdominal muscles and lifting of the abdomen off the platform (3); arching of the body and lifting of the pelvic structures and scrotum (4). 

### 4.5. Histological Evaluation of Colon Damage

The evaluation of colon damage was performed at the macroscopic level in accordance with the criteria previously reported [33]. The macroscopic criteria were: the presence of adhesions between the colon and other intra-abdominal organs (0–2); consistency of colonic faecal material (an indirect marker of diarrhoea; 0–2); thickening of the colonic wall (mm); presence and extension of hyperaemia and macroscopic mucosal damage (0–5). Colon length was measured after the explant of the tissue. For the histological analysis, the colon was fixed in 4% paraformaldehyde for 24 h, dehydrated in alcohol, included in paraffin, and cut into 5 µm sections, by using the “Swiss roll” technique to visualize all the colon length in each tissue section. Microscopic evaluations of colon damage (mucosal architecture loss, cellular infiltrate, muscle thickening, crypt abscess, and goblet cell depletion) were carried out on haematoxylin/eosin-stained sections. The infiltration of mast cells (MCs) and eosinophils was investigated on colon sections stained with GIEMSA (Sigma-Aldrich, Milan, Italy). Digitalized images were collected by a Leica DMRB light microscope equipped with a DFC480 digital camera (40× magnification; Leica Microsystems, Wetzlar, Germany). The quantitative analysis was carried out by two blind investigators with the software ImageJ. For each animal, the cellular density (cell number/respective arbitrary field) of 3–5 independent arbitrary optical fields (0.1 mm^2^) collected from the submucosa was measured. 

### 4.6. Immunofluorescence Analysis

For immunoreactions, the colon was cut into 5 µm slices (by using the “Swiss roll” technique to have all the colon length in each tissue section) and dried on glass slides prior to deparaffinization with xylol and rehydration in a descending alcohol series (100, 95, 75, and 50%). Tissues were rinsed three times (5 min each) in PBS containing 0.1% Triton X-100 (T-PBS) followed by a 1 h incubation in blocking solution (containing 0.1% Triton X-100, and 5% bovine serum albumin in 1X PBS) at room temperature. The slices were incubated overnight at 4 °C with a mouse anti-UCH-L1/PGP9.5 (Novus Biologicals-31A3, Bio-Techne Ltd., Abingdon, UK), diluted 1:500 in T-PBS/5% BSA (Sigma-Aldrich, Milan, Italy) and a rabbit anti-glial fibrillary acidic protein (GFAP, DAKO-Z0334, Agilent Technologies Italia, Milan, Italy), diluted 1:500 in PBS/5% BSA (Sigma-Aldrich, Milan, Italy). The following day, slides were washed thrice with PBS and then incubated in blocking solution for 1 h with goat anti-mouse (Invitrogen-Thermo Fisher Scientific, Milan, Italy) and goat anti-rabbit (Invitrogen-Thermo Fisher Scientific, Milan, Italy) secondary antibodies labelled with Alexa Fluor 488 and 647, respectively. To stain the nuclei, sections were incubated with DAPI in PBS for 5 min at room temperature in the dark. After three washes in PBS and a final wash in distilled water, slices were mounted using Fluoromount-G™ Mounting Medium (Thermo Fisher Scientific, Milan, Italy) as mounting medium. After the paraformaldehyde (4%) fixation and the snap freezing, the lumbosacral segment of the spinal cord was cut into 5 µm slices, rinsed three times (5 min each) in PBS containing 0.1% Triton X-100 (T-PBS), followed by a 1 h incubation in blocking solution (containing 0.1% Triton X-100, and 5% bovine serum albumin in 1X PBS) at room temperature. The slices were incubated overnight at 4 °C with a rabbit anti-Iba1 (rabbit antiserum, 1:500; Wako Chemicals, Richmond, VA, USA), diluted 1:250 in T-PBS/5% BSA (Sigma-Aldrich, Milan, Italy) or with a rabbit anti-glial fibrillary acidic protein (GFAP, DAKO-Z0334, Agilent Technologies Italia, Milan, Italy), diluted 1:500 in PBS/5% BSA (Sigma-Aldrich, Milan, Italy). The following day, slides were washed thrice with PBS and then incubated in blocking solution for 1 h with a goat anti-rabbit (Invitrogen-Thermo Fisher Scientific, Milan, Italy) secondary antibodies labelled with Alexa Fluor 647 and 488, respectively. To stain the nuclei, sections were incubated with DAPI in PBS for 5 min at room temperature in the dark. After three washes in PBS and a final wash in distilled water, slices were mounted using Fluoromount-G™ Mounting Medium (Thermo Fisher Scientific, Milan, Italy) as mounting medium.

Digitalized images were collected at 400× (colon) and 200× (spinal cord) total magnification using a motorized Leica microscope DM6 B equipped with a DFC9000 GT camera, supported by a THUNDER Workstation 3D DCV and by the software LAS X (Leica Biosystems, Milan, Italy). The quantitative analysis of colon PGP9.5- and GFAP-related immunofluorescence intensity was performed by collecting independent fields (4–6 for each animal) from the myenteric plexus and analysing them with FIJI software (NIH, Bethesda, MD, USA). The number of PGP9.5- and GFAP-positive cells for each myenteric plexus was counted and normalized to the plexus area. The immunofluorescence relative to the expression of PGP9.5 and GFAP was quantified (arbitrary units), normalized to the area of the myenteric plexus, and results were expressed as a percentage of the control group. The value relative to the background was subtracted from the value obtained from the analysed area, and the results were expressed as a percentage of the control group. Quantitative analysis of GFAP- and Iba1-positive cells was performed by collecting at least three independent fields through a ×20 0.5NA objective. GFAP-positive cells were counted using the “cell counter” plugin of ImageJ, whereas Iba1-positive cells were quantified using the automatic thresholding and segmentation features of ImageJ. Quantification of GFAP signal in immune-stained sections was also performed using FIJI software by automatic thresholding images with the aid of an algorithm, which delivered the most consistent pattern recognition across all acquired images.

### 4.7. Statistical Analysis

Behavioural measurements were performed on 10 animals for each treatment. Histological analyses were performed on five animals for each treatment. All the experimental procedures were performed by a researcher blind to the treatment. Results were expressed as mean ± S.E.M. The analysis of variance of the data was performed by one-way ANOVA with Bonferroni’s significant difference procedure used for post-hoc comparisons. Statistics of electrophysiological data were performed by Student’s paired or unpaired t-test or by one-way ANOVA, followed by Bonferroni’s post-test, as appropriate. *p* values of less than 0.05 were considered significant. Data were analysed using the “Origin 9” software (OriginLab, Northampton, MA, USA).

## Figures and Tables

**Figure 1 ijms-24-14841-f001:**
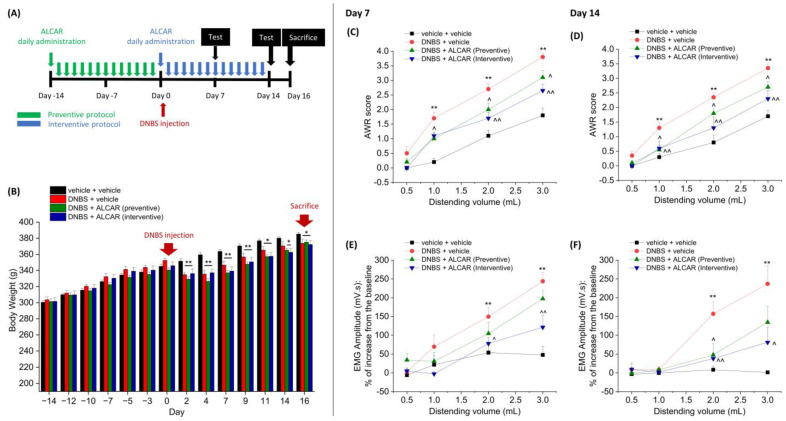
Effect of the repeated administration of ALCAR on visceral pain induced by DNBS in rats. ALCAR (100 mg kg^−1^ s.c.) was administered twice daily in the DNBS-treated animals, according to the preventive (green) or the interventive protocol (light blue), respectively (**A**). Body weight (g) has been monitored over the course of the experiment (**B**). Visceral sensitivity was assessed by measuring the extent of the abdominal withdrawal response (AWR; (**C**,**D**)) and the visceromotor response (VMR; (**E**,**F**)) to colorectal distension (0.5–3 mL). Tests were performed on days 7 (acute inflammatory phase; (**C**,**E**)) and 14 (post-inflammatory phase; (**D**,**F**)), 24 h after the last administration. Each value represents the mean ± SEM of 10 animals per group. * *p* <0.05 and ** *p* <0.01 vs. vehicle + vehicle treated animals. ^ *p* < 0.05 and ^^ *p* < 0.01 vs. DNBS + vehicle treated animals.

**Figure 2 ijms-24-14841-f002:**
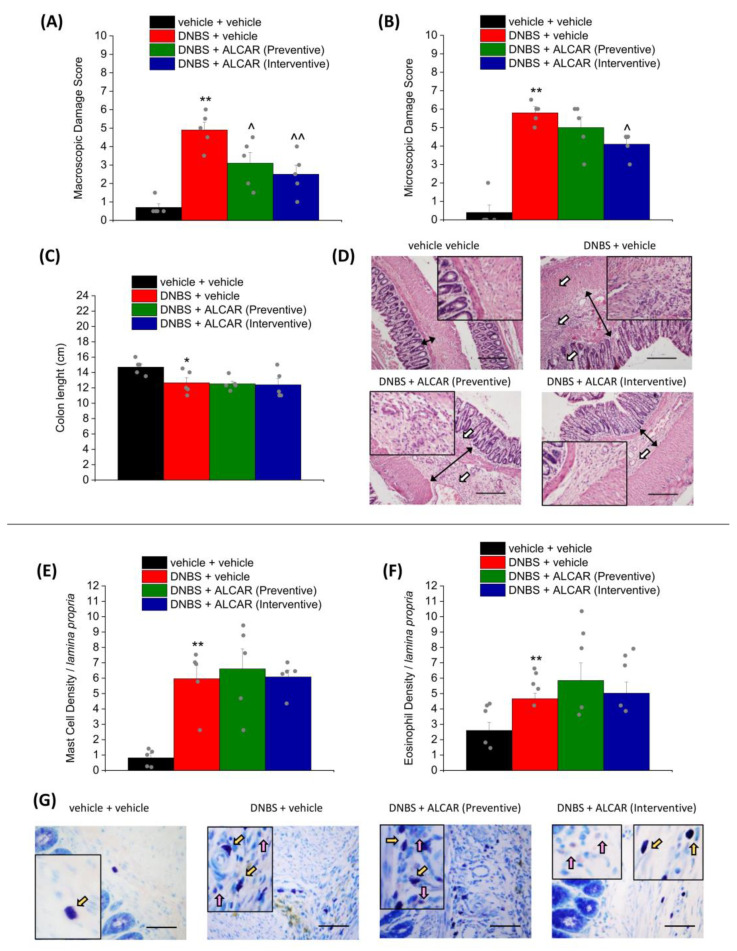
The effects of repeated treatment with ALCAR on colon damage induced by DNBS in rats. ALCAR (100 mg kg^−1^ s.c.) was administered twice daily in the DNBS-treated animals, according to the preventive (green) and the interventive protocol (light blue), respectively; then tissues were collected (Day 16, after the end of behavioural tests). The column graphs report the colon macroscopic (**A**) and microscopic (**B**) damage score and the colon length (**C**); Representative pictures of haematoxylin–eosin-stained sections of full-thickness colon (**D**); Original magnification 10×; scale bar 200 μm) show the infiltrate (white arrows) and submucosa thickness (black double arrows). The column graph displays the mean mast cell density (**E**) and the mean eosinophil density (**F**) per area of the colonic wall (number of cells/0.05 mm^2^ lamina propria). The panel (**G**) shows pictures captured from submucosa of mast cell granules stained in purple (yellow arrows) and eosinophils stained in pink arrows) with GIEMSA (Original magnification 40×; scale bar 50 μm). Microscopic evaluations of colon damage were carried out on haematoxylin/eosin-stained sections (4 for each animal). Five independent arbitrary optical fields (0.1 mm^2^) from the submucosa of 4 sections of the colon for each animal were collected and used for the analysis of cell density. Each value represents the mean ± SEM of 5 animals per group (the grey dots represent the values of individual animals within each group). * *p* < 0.05 and ** *p* < 0.01 vs. vehicle + vehicle. ^ *p* < 0.05 and ^^ *p* < 0.01 vs. DNBS + vehicle.

**Figure 3 ijms-24-14841-f003:**
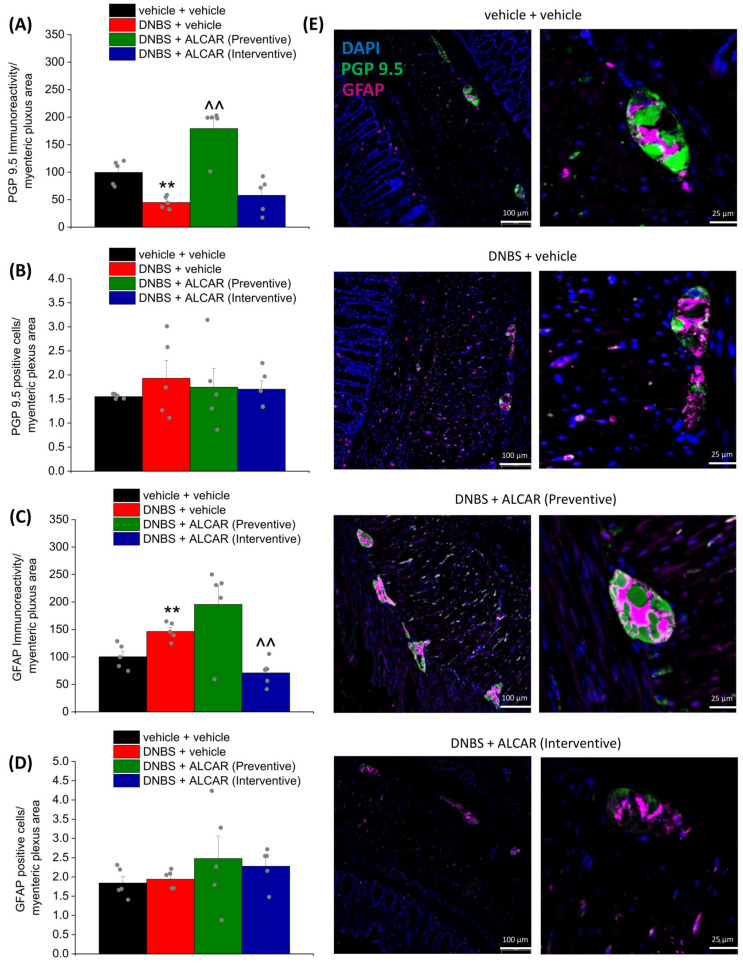
The neuroprotective effects of ALCAR on enteric nervous system damage caused by DNBS. ALCAR (100 mg kg^−1^ s.c.) was administered twice daily in the DNBS-treated animals, according to the preventive (green) and the interventive protocol (light blue), respectively; tissues were collected on Day 16, after the end of behavioural tests. The figure reports the immunolabeling quantification of PGP 9.5 (**A**), the density of PGP 9.5 positive cells (**B**), the immunolabeling quantification of GFAP (**C**), and the density of GFAP positive cells (**D**) with relative immunofluorescence images showing the expression of PGP 9.5 (green), GFAP (purple), and DAPI (blue) in the myenteric plexus of the colon (**E**). The quantitative analysis of PGP9.5- and GFAP-related immunofluorescence intensity (arbitrary unit) was performed by collecting independent fields (4–6 for each animal) from the colon (3–4 sections for each animal) and by analysing the area related to the myenteric plexus. Immunoreactivity was expressed as a percentage of the control group (vehicle + vehicle-treated animals). Each value represents the mean ± SEM of 5 animals per group (the grey dots represent the values of individual animals within each group). ** *p* < 0.01 vs. vehicle + vehicle. ^^ *p* < 0.01 vs. DNBS + vehicle. Original magnification: 20×.

**Figure 4 ijms-24-14841-f004:**
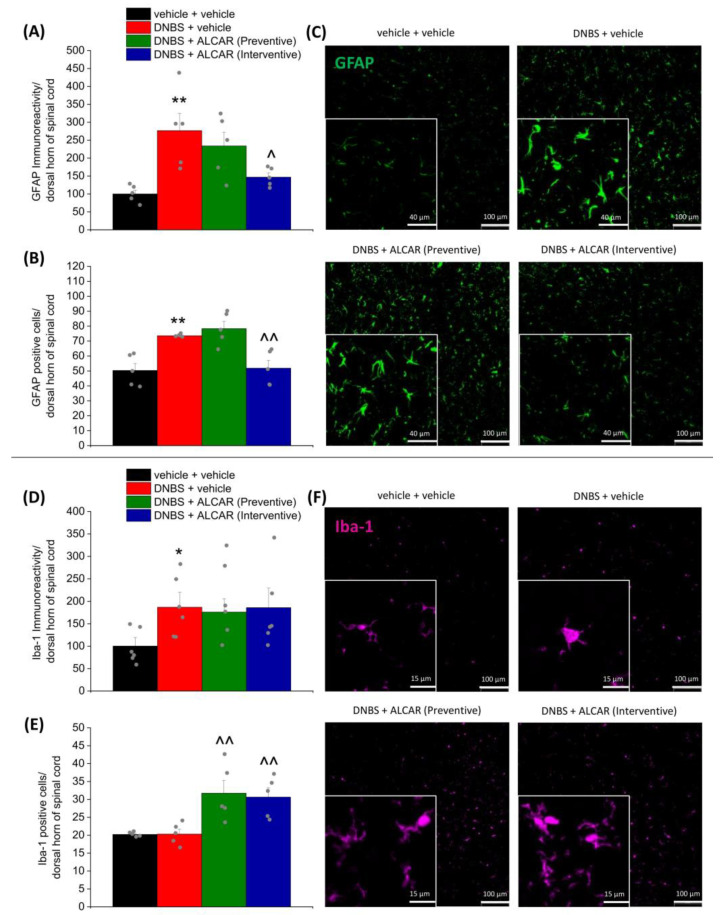
The effects of ALCAR on glia overactivation in the spinal cord of DNBS-treated animals. ALCAR (100 mg kg^−1^ s.c.) was administered twice daily in the DNBS-treated animals, according to the preventive (green) and the interventive protocol (light blue), respectively; tissues were collected on Day 16, after the end of behavioural tests. The figure reports the immunolabeling quantification of GFAP (**A**), the density of GFAP-positive cells (**B**), the immunolabeling quantification of Iba-1 (**D**), and the density of Iba-1-positive cells (**E**) with relative immunofluorescence images showing the expression of GFAP (green; (**C**)) and Iba-1 (purple; (**F**)). The quantitative analysis of GFAP- and Iba-1-related immunofluorescence intensity (arbitrary unit) was performed by collecting independent fields (4–6 for each animal) from the dorsal horns of the spinal cord (4–6 sections for each animal). Immunoreactivity was expressed as a percentage of the control group (vehicle + vehicle-treated animals). Each value represents the mean ± SEM of 5 animals per group (the grey dots represent the values of individual animals within each group). * *p* < 0.05, ** *p* < 0.01 vs. vehicle + vehicle. ^ *p* < 0.05, and ^^ *p* < 0.01 vs. DNBS + vehicle. Original magnification: 20×.

## Data Availability

The data presented in this study are available on request from the corresponding author. The data are not publicly available due to privacy restrictions.

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
