# Peer review of "Anti-Hyperalgesic Efficacy of Acetyl L-Carnitine (ALCAR) Against Visceral Pain Induced by Colitis: Involvement of Glia in the Enteric and Central Nervous System"

_ijms, 2023, doi:10.3390/ijms241914841_

Round 1
Reviewer 1 Report
In the manuscript ijms-2476728, titled as “Anti-hyperalgesic efficacy of acetyl l-carnitine (ALCAR) against visceral pain induced by colitis: involvement of glia in enteric and central nervous system” by Lucarini and coworkers, the authors report the anti-hyperalgesic efficacy of Acetyl L-carnitine in a model of persistent visceral pain associated with colitis induced by 2,4-dinitrobenzenesulfonic acid intrarectal injection. Since to date, there were no evidence in literature attesting ALCAR efficacy on pain of gastrointestinal origin, results of ALCAR sound very interesting.
Although this paper is adding only a small piece of evidence to what it is already known on ALCAR roles, it adds some novel information supporting and confirming previous findings. The manuscript is well written and the experiments and data support the authors conclusions. The data are novel, clear and important for the understanding of the pathology of gastro-intestinal symptoms.
I have some important comments, which might further improve the authors work:
-since the authors well explained the possible mechanism by which ALCAR reduce the GI sensitivity, i.e the regulation of M1 muscarinic receptors and or mGlu2R activity, why they did not measure the protein expression in the GI system? ALCAR exhibits many different biological actions. It would have been important to interfere with some receptors (M1 receptor or mGlu2) to demonstrated a possible mechanism, as the authors did in their previous papers (PMID: 12504925). If this is not possible I would clearly state it in the discussion.
- may the authors clarify why they use the preventive and intervention protocols and they did no try to administer ALCAR for the entire period (preventive + intervention). Since ALCAR display a slow effect/mechanism it would be interesting to evaluate this paradigm. The authors should anyway clarify in the methods section why they used such protocol;
- figure 2: please display the scale bar in the legend (C and F);
- 156-158: what does “the suffering of enteric neurons” mean? I am also not sure whether PGP 9.5 is a good marker for proving the degenerative status of these neurons (see also 313-315). I would use another marker or tone down the phrase;
- 165-166: From the images is not easy to see an increased immunoreactivity of the astrocytic marker GFAP. Moreover, the PGP 9.5 signal partially overlaps with the GFAP, which is not possible. I recommend to use high quality images and also display them with higher magnification;
- figure 3E: scale bar missing;
- figure 4C and F: scale bar missing;
- 197-200: here is confusing, when compared to the related images. I did not understand how the percentage may change if the density is stable. Did the total area change? Moreover, it is not clear if the authors are referring to the D or E panel. Please, rephrase the sentence;
- 4F: the magnification is very low and is impossible to draw a conclusion. If the authors what to demonstrate a significant change in the microglia morphology, they should show it clearly.
-I would mention the entire name 2,4-dinitrobenzenesulfonic acid (DNBS) even in
Author Response
Dear Editor and Reviewers,
We would like to thank you for your efforts in reviewing our manuscript titled “Anti-hyperalgesic efficacy of acetyl l-carnitine (ALCAR) against visceral pain induced by colitis: involvement of glia in enteric and central nervous system” providing helpful comments and suggestions. We have studied your comments and revised the manuscript accordingly. All authors have approved the response letter and the revised version of the manuscript.
Reviewer 1
Comments and Suggestions for Authors
In the manuscript ijms-2476728, titled as “Anti-hyperalgesic efficacy of acetyl l-carnitine (ALCAR) against visceral pain induced by colitis: involvement of glia in enteric and central nervous system” by Lucarini and coworkers, the authors report the anti-hyperalgesic efficacy of Acetyl L-carnitine in a model of persistent visceral pain associated with colitis induced by 2,4-dinitrobenzenesulfonic acid intrarectal injection. Since to date, there were no evidence in literature attesting ALCAR efficacy on pain of gastrointestinal origin, results of ALCAR sound very interesting.
Although this paper is adding only a small piece of evidence to what it is already known on ALCAR roles, it adds some novel information supporting and confirming previous findings. The manuscript is well written and the experiments and data support the authors conclusions. The data are novel, clear and important for the understanding of the pathology of gastro-intestinal symptoms.
Response. Thanks for appreciating the manuscript.
I have some important comments, which might further improve the authors work:
-since the authors well explained the possible mechanism by which ALCAR reduce the GI sensitivity, i.e the regulation of M1 muscarinic receptors and or mGlu2R activity, why they did not measure the protein expression in the GI system? ALCAR exhibits many different biological actions. It would have been important to interfere with some receptors (M1 receptor or mGlu2) to demonstrated a possible mechanism, as the authors did in their previous papers (PMID: 12504925). If this is not possible, I would clearly state it in the discussion.
Response. Thanks for the comment. We discussed some mechanisms which might explain ALCAR efficacy against visceral pain based on the evidence in literature. ALCAR is involved in metabolism, as in several other signalling pathways, regulating the expression of many proteins as well as the activity of receptors involved in pain physiology. The pleiotropic activity of ALCAR represents a strength in the therapy of visceral pain, which displays a complex nature. At the same time, it makes difficult to study ALCAR mechanism in specific contexts, since the effect of the repeated administration with ALCAR might be the results of several concomitant changes: ALCAR might act horizontally on different targets, but also vertically on different partners of the same pathway. The results we can obtain by investigating a single factor could led to misleading conclusions. On the other hand, changes in the expression of receptors/proteins (already demonstrated in literature) will not definitely demonstrate or exclude any mechanisms beyond the effect of ALCAR. Yet, demonstrating the mechanisms would require dedicated functional studies which, based on the evidence collected in this work, might feed into future researches. According to the reviewer suggestion, we stated it in the discussion.
- may the authors clarify why they use the preventive and intervention protocols and they did no try to administer ALCAR for the entire period (preventive + intervention). Since ALCAR display a slow effect/mechanism it would be interesting to evaluate this paradigm. The authors should anyway clarify in the methods section why they used such protocol;
Response. Thanks for the comment. We agree with the Reviewer that might be interesting to evaluate the effect of administering ALCAR for the entire period (preventive + intervention), also in the view to enhance the efficacy of the treatment. In the present work we decided to evaluate the effect of the preventive and interventive protocols separately, to highlight the long-term protective efficacy of ALCAR against the development of pain, which further support its employment as a maintenance therapy in IBD patients. Indeed, the pre-treatment with ALCAR provided a slight but significant protection of the colon from the inflammatory insults, attenuating the tissue damage as well as pain. The obtained results proved that some of the beneficial effect of ALCAR on pain associated with colitis are completely independent of the regulation of inflammatory process, which represents a very innovative aspect to further explore in the view to develop a selective therapy for pain. If we had combined the preventive protocol with the interventive, these properties would not have emerged. We clarified this point in the manuscript, according to Reviewer suggestion.
- figure 2: please display the scale bar in the legend (C and F);
Response. Thanks for the comment. We introduced the scale bar in the legend.
- 156-158: what does “the suffering of enteric neurons” mean? I am also not sure whether PGP 9.5 is a good marker for proving the degenerative status of these neurons (see also 313-315). I would use another marker or tone down the phrase;
Response. Thanks for the comment. To the best of our knowledge, a reduced expression of PGP 9.5 has been associated with axonopathy, neuropathy or enteric neurons damage (doi: 10.1016/j.pneurobio.2009.10.020; doi: 10.18632/aging.101677; doi: 10.1186/s12906-017-1847-4; doi: 10.3389/fpain.2021.731658; doi: 10.1111/jcmm.12428). Anyway, we revised the sentences according to the Reviewer suggestion.
- 165-166: From the images is not easy to see an increased immunoreactivity of the astrocytic marker GFAP. Moreover, the PGP 9.5 signal partially overlaps with the GFAP, which is not possible. I recommend to use high quality images and also display them with higher magnification;
Response. Thanks for the comment. We revised the panel to make the images easy to see. We also added inserts with higher magnification.
- figure 3E: scale bar missing;
Response. Scale bar is present in the images, but it is difficult to see because of the images size. We revised the figure to make the scale bar easy to see.
- figure 4C and F: scale bar missing;
Response. Scale bar is present in the images, but it is difficult to see because of the images size. We revised the figure to make the scale bar easy to see.
- 197-200: here is confusing, when compared to the related images. I did not understand how the percentage may change if the density is stable. Did the total area change? Moreover, it is not clear if the authors are referring to the D or E panel. Please, rephrase the sentence;
Response. Thanks for the comment. The immunoreactivity related to Iba-1 was 2.5-fold higher in DNBS+vehicle treated animals, as compared to controls. This difference was not attributable to an increase in the Iba1-positive cells density, but to the morpho-functional changes that the microglia underwent (as the enlargement of the body size and the increase in the expression of the marker). We indicated within the brackets the respective reference to immunoreactivity quantification (D), density of Iba-1 positive cells (E) and the representative images of microglia morphology (F). We added inserts with higher magnification to better highlight microglia morphological changes. We also revised the manuscript in order to clarify this point.
- 4F: the magnification is very low and is impossible to draw a conclusion. If the authors what to demonstrate a significant change in the microglia morphology, they should show it clearly.
Response. Thanks for the suggestion. We added inserts in the images to show clearly the changes in microglia morphology.
-I would mention the entire name 2,4-dinitrobenzenesulfonic acid (DNBS) even in
Response. Thanks for the suggestion. We revised the text accordingly.

Reviewer 2 Report
The topic of the manuscript is interesting, but it must be much improved in many aspects before publication.
Here are the comments:
- Authors use a DNBS induced colitis model, they must explain in more detail at section 4.2. I guess that DNBS administration in done daily during 14 days but this information must be explicitly indicated.
- Authors should indicate if the colitis induced with this model is acute or chronic, they say (line 96) that at day 8 treated rats are in the acute phase and at day 15 the colitis is resolved (line 97). Actually, I do not think that only one day after the last DNBS dose the colitis will be resolved. Authors must abord this idea and justify why all the histological and immunofluorescence assays are done at day 15.
- Why have authors not done all the study also at day 8? Authors must justify.
- I think that Figure 1 should show the protocol (clearer than it is shown now), and all the information that inform us about the severity of the colitis, e.g. H&E stainings, macroscopic and microscopic damages and eosinophiles and mast cells infiltration. And all this information also at day 8 of treatment. Then, the rest of the figures.
- Figure 2:
§ Indicate scale bar length (2B and 2F).
§ Use arrows to point the areas where the damage is seen, e.g, muscular thick, the infiltration in submucosa…(2C)
§ 2C: in the DNBS+ALCAR (interventive) condition: the thick of the muscular layer is quite big also, however authors say that the layer was significantly less thick (line 131).
§ 2D and 2E: the legend of y axis is not clear, indicate something like “number of mast cells / xxx mm2 lamina propria”
§ 2B and 2F: show a lower magnification image and then a higher magnification image as an insert highlighting the effect that is considered important. Use also arrows.
§ 2F: I cannot see the red color.
§ Figure legend: the last sentence is repeated.
- Paragraph in lines 163-171: it is a mess. The positions of all the letters are wrong.
- Line 166: you say that “GFAP is a peculiar marker of enteric glia activation”, actually GFAP is an specific marker of astrocytes (not of all glia activation, is a marker of astrocytes in enteric or central nervous system, not only enteric).
- Figure 3:
§ Why authors indicate in the legend (3A-D): vehicle+vehicle?
§ 3E: indicate scale bars and their length.
§ Show bigger images, Now is really hard to see anything, and improve the quality of the images.
§ Legend: there are lots of mistakes, also in statistics.
§ Explain the method for determine PGP positive neurons in method section, now is missed.
- Section 2.4: if authors want to show microglia activations the must quantify the change in microglia morphology. What they have shown now (the immunofluorescence intensity) is not a marker of microglia activation as they say at line 199.
- Figure 4:
§ Also in the legends: “vehicle+vehicle”
§ Show scales bars and the length
§ The signification of “*” is missed in the legend
- Discussion:
§ The authors discuss for a long paragraph (lines 227-258) the possible mechanisms of ALCAR by increasing the expression of different receptors such as mGlu2R or M1 muscarinic receptor, however they do not measured the expression of none of them. I wonder why authors do not measure the expression of any of these receptors, maybe by real-time PCR, in order to complete the study. I think this would be positive for the manuscript.
§ Author talk about Astrogliosis in line 301, for the first time but they do not explain nothing about that.
§ Authors have not demonstrate a microglia activation in spinal cord , for doing that they must to measure morphological changes in microglia.
§ Authors actually cannot explain if the increase in microglia density in spinal cord is a positive or negative effect, because microglia activation can results in an increase in the production of pro-inflammatory or anti-inflammatory molecules, maybe you could measure some pro or anti-inflammatory cytokines by real-time PCR in spinal cord.
§ With these results, conclusions given in lines 317-320 cannot be done.
English quality is average
Author Response
Dear Editor and Reviewers,
We would like to thank you for your efforts in reviewing our manuscript titled “Anti-hyperalgesic efficacy of acetyl l-carnitine (ALCAR) against visceral pain induced by colitis: involvement of glia in enteric and central nervous system” providing helpful comments and suggestions. We have studied your comments and revised the manuscript accordingly. All authors have approved the response letter and the revised version of the manuscript.
Reviewer 2
Comments and Suggestions for Authors
The topic of the manuscript is interesting, but it must be much improved in many aspects before publication.
Response. We are grateful to the Reviewer comments, which contributed much to the improvement of the manuscript.
Here are the comments:
- Authors use a DNBS induced colitis model, they must explain in more detail at section 4.2. I guess that DNBS administration in done daily during 14 days but this information must be explicitly indicated.
Response. We apologize for the lack of clarity. The model of colitis induced by DNBS injection provides for a single injection on this sensitizing agent on day 0 as correctly represented in Figure 1A. We specifically indicated it in the methods section, according to reviewer suggestion.
- Authors should indicate if the colitis induced with this model is acute or chronic, they say (line 96) that at day 8 treated rats are in the acute phase and at day 15 the colitis is resolved (line 97). Actually, I do not think that only one day after the last DNBS dose the colitis will be resolved. Authors must abord this idea and justify why all the histological and immunofluorescence assays are done at day 15.
Response. Thanks for the comment. As documented in literature, a single injection of DNBS is enough to establish a model of colitis in the animals. This type of colitis has a peak between 3 and 7 days after DNBS injection, then colitis gradually goes into remission. On day 8 animals are in the acute phase of colitis. On day 15 instead the damage is significantly reduced with respect of the acute one, in this phase pain persists even if the colon is healing. Literature identifies this stage as post-inflammatory and recognize it as a key point for the turning of pain from acute to chronic (doi: 10.3390/cells9081772). A large amount of IBD patients continue to manifest abdominal pain in the remission phase of the disease. Indeed, even if anti-inflammatory, immunomodulant or disease-modifying agents are effective in treating the inflammatory pathology, they often do not prevent the establishment of chronic abdominal pain in these patients. Unfortunately, this kind of chronic pain is still considered untreatable, because it is not responsive to the classical pain-killers (doi: 10.1093/crocol/otab073). Since the aim of our study was to evaluate the anti-hyperalgesic efficacy of ALCAR against the persistent pain resulting from the colon damage, we focused our histological analysis on day 15. Indeed, we were interested in understanding to what the anti-hyperalgesic efficacy of ALCAR was associated with, a protection from colon damage or a neuromodulator effect. From this data it emerged that treatment with ALCAR is more effective in counteracting visceral pain and in preserving nervous tissue than in preventing intestinal inflammation.
- Why have authors not done all the study also at day 8? Authors must justify.
Response. As anticipated in the previous comment, we were particularly interested in the effect of ALCAR on visceral pain establishment and persistence, so we focused our study on the post-inflammatory phase of colitis (day 15). In this context, we investigated the impact of the treatment on some mechanisms involved in the pathophysiology of chronic intestinal hypersensitivity, such as mast cell infiltration and dysfunction, enteric neuroplasticity, enteric glia and spinal glia overactivation (doi: 10.3390/cells9081772; doi: 10.3390/biomedicines9111671; doi: 10.1111/nmo.14339; doi: 10.1053/j.gastro.2022.02.016; doi.org/10.1038/nrgastro.2014.103). These factors take on a key role in the persistence of visceral pain beyond colon tissue healing, which represent our clinical target. We pointed out this aspect in the manuscript (“Methods” section).
- I think that Figure 1 should show the protocol (clearer than it is shown now), and all the information that inform us about the severity of the colitis, e.g. H&E stainings, macroscopic and microscopic damages and eosinophiles and mast cells infiltration. And all this information also at day 8 of treatment. Then, the rest of the figures.
Response. Thank for the suggestions. We revised the scheme of the protocols in order to make it clearer than now (the timeline was revised, accordingly). Regarding the severity of colitis, as previously discussed, we choose to focus our study on the post-inflammatory phase, performing the analysis on day 15. In the acute phase of colitis, we could deepen mainly the anti-inflammatory potential of the treatment, which is a side aspect of our project. Indeed, as previously mentioned and published, inflammation and pain follow different courses in animals like in patients (doi: 10.3390/cells9081772; doi: 10.1093/crocol/otab073). According to our objectives, in Figure 1, we choose to describe the effect of ALCAR on pain establishment and persistence, highlighting and comparing the effect of both the protocols of treatment. We retain more adequate to show the colon histology in a separate Figure, otherwise the images would result too small to appreciate differences and similarities.
- Figure 2:
- Indicate scale bar length (2B and 2F).
- Use arrows to point the areas where the damage is seen, e.g, muscular thick, the infiltration in submucosa…(2C)
- 2C: in the DNBS+ALCAR (interventive) condition: the thick of the muscular layer is quite big also, however authors say that the layer was significantly less thick (line 131).
- 2D and 2E: the legend of y axis is not clear, indicate something like “number of mast cells / xxx mm2 lamina propria”
- 2B and 2F: show a lower magnification image and then a higher magnification image as an insert highlighting the effect that is considered important. Use also arrows.
- 2F: I cannot see the red color.
- Figure legend: the last sentence is repeated.
Response. Thanks for the suggestions. We revised the figures and the legends according to Reviewer comments. The submucosa layer in the colon of DNBS+ALCAR (interventive) condition is less thick than DNBS + vehicle. We specified it in the results.
- Paragraph in lines 163-171: it is a mess. The positions of all the letters are wrong.
Response. We apologize for the confusion in the references. We checked and revised the paragraph accordingly.
- Line 166: you say that “GFAP is a peculiar marker of enteric glia activation”, actually GFAP is an specific marker of astrocytes (not of all glia activation, is a marker of astrocytes in enteric or central nervous system, not only enteric).
Response. We partially agree with the comment of the Reviewer. Indeed, our sentence was referred to the intestinal context where actually GFAP can be considered a peculiar marker of enteric glia, since there are no astrocytes there, neither other cellular type expressing GFAP (doi: 10.1113/JP271021). An upregulation of GFAP expression can be detected in reactive astrocytes in the central nervous system, as well as in reactive enteric glial cells in the gastrointestinal tract (doi: 10.3390/biomedicines9111671; doi: 10.3390/cells9081772, doi: 10.1136/gut.2003.012625), since these cells share several phenotypic characteristics (doi: 10.1113/JP271021).
- Figure 3:
- Why authors indicate in the legend (3A-D): vehicle+vehicle?
- 3E: indicate scale bars and their length.
- Show bigger images, Now is really hard to see anything, and improve the quality of the images.
- Legend: there are lots of mistakes, also in statistics.
- Explain the method for determine PGP positive neurons in method section, now is missed.
Response. Thanks for the comments. We revised the Figure, in particular we made the images bigger so their quality can be appreciated, we revised the legend, and we added the missing information to the methods section. The nomenclature “vehicle + vehicle” is the correct one”. There was a mistake in the nomenclature of Figure 2.
- Section 2.4: if authors want to show microglia activations the must quantify the change in microglia morphology. What they have shown now (the immunofluorescence intensity) is not a marker of microglia activation as they say at line 199.
Response. Thanks for the comment. As correctly pointed out by the reviewer, the change in activation status of microglia is reflected in its gradual morphological transformation from a highly ramified into a less ramified or amoeboid cell shape. Iba1 is a cytoplasmic calcium-binding protein that is expressed in cells of monocytic lineage and immunolabeling of Iba1 in the cytoskeleton of microglia allows for quantification of changes in morphology. Indeed, in the CNS, Iba1 is expressed in microglial cells and is upregulated during microglia activation. Moreover, when activated, microglia undergo rapid proliferation. Therefore, a crude way of identifying an overactivation of microglia is purely by counting the number of microglial cells or measuring Iba1 expression (as assessed by the quantification of Iba-1 immunoreactivity). A microgliosis is always a sign of ongoing pathology that is caused by microglial cell proliferation and/or myeloid cell infiltration. Unless a developing brain with a physiological microgliosis is examined, the observation of higher numbers of microglia cells alone proves that the tissue is not completely homeostatic or healthy (doi: 10.1007/s00401-021-02370-8). Anyway, we understand the concern of the Reviewer about the low resolution of images and the impossibility of appreciating microglia morphology, so we added inserts with a higher magnification and resolution within each representative image.
- Figure 4:
- Also in the legends: “vehicle+vehicle”
- Show scales bars and the length
- The signification of “*” is missed in the legend
Response. Thanks for the comments. We revised the Figure and added the missing information.
- Discussion:
- The authors discuss for a long paragraph (lines 227-258) the possible mechanisms of ALCAR by increasing the expression of different receptors such as mGlu2R or M1 muscarinic receptor, however they do not measured the expression of none of them. I wonder why authors do not measure the expression of any of these receptors, maybe by real-time PCR, in order to complete the study. I think this would be positive for the manuscript.
Response. Thanks for the comment. We discussed some mechanisms which might explain ALCAR efficacy against visceral pain based on the evidence in literature. ALCAR is involved in metabolism, as in several other signalling pathways, regulating the expression of several proteins as well as the activity of receptors involved in pain physiology. The pleiotropic activity of ALCAR represents a strength in the therapy of visceral pain, which displays a complex nature. At the same time, it makes difficult to study ALCAR mechanism in specific contexts, since the effect of the repeated administration with ALCAR might be the results of several concomitant changes: ALCAR might act horizontally on different targets, but also vertically on different partners of the same pathway. The results we can obtain by investigating a single factor could led to misleading conclusions. On the other hand, changes in the expression of receptors/proteins (already demonstrated in literature) will not definitely demonstrate or exclude any mechanisms beyond the effect of ALCAR. Yet, demonstrating the mechanisms would require dedicated functional studies which, based on the evidence collected in this work, might feed into future researches. We stated it in the discussion.
- Author talk about Astrogliosis in line 301, for the first time but they do not explain nothing about that.
Response. Thanks for the comment. We deepen astrogliosis topic in the revised version of the manuscript.
- Authors have not demonstrate a microglia activation in spinal cord , for doing that they must to measure morphological changes in microglia.
- Authors actually cannot explain if the increase in microglia density in spinal cord is a positive or negative effect, because microglia activation can results in an increase in the production of pro-inflammatory or anti-inflammatory molecules, maybe you could measure some pro or anti-inflammatory cytokines by real-time PCR in spinal cord.
Response. Thanks for the comments. As we discussed in the previous comments, Iba1 is upregulated during microglia activation. Moreover, when activated, microglia undergo rapid proliferation. Therefore, a crude way of identifying an overactivation of microglia is purely by counting the number of microglial cells or measuring Iba1 expression (as assessed by the quantification of Iba-1 immunoreactivity). A microgliosis is always a sign of ongoing pathology, even though microglia can assume a positive (resolutive) or a negative (neurodegenerative) effect. Indeed, unless a developing brain with a physiological microgliosis is examined, the observation of higher numbers of microglia cells alone proves that the tissue is not completely homeostatic or healthy (doi: 10.1007/s00401-021-02370-8). On the other hand, we think it is important to underlie that comparisons of quantitative approaches for assessing microglial morphology are often inconsistent and still need for standardization (doi: 10.1038/s41598-022-23091-2). In addition, while conventional imaging methods suggest that microglia form a comparatively homogenous population, recent studies demonstrated that microglia are heterogeneous. By using single-cell RNA-sequencing technology, microglia can be divided into distinct (disease-specific) states or subclusters (doi: 10.3390/cells11152383; doi: 10.1002/iid3.362; doi: 10.1016/j.celrep.2020.01.010). Therefore, only the gene expression pattern of the disease-associated clusters might properly describe the functional changes occurring in the activated microglial cells more than cytokines expression. Indeed, during inflammatory states there is a continuous balance between pro-inflammatory and anti-inflammatory cytokines which cannot give a direct indication of microglia function in the context.
- With these results, conclusions given in lines 317-320 cannot be done.
Response. According to Reviewer comment, we revised the paragraph to avoid overstatements.

Round 2
Reviewer 1 Report
This reviewer is satisfied with the authors' responses.
Author Response
Dear Reviewer,
We would like to thank you for approving the revision of our manuscript titled “Anti-hyperalgesic efficacy of acetyl l-carnitine (ALCAR) against visceral pain induced by colitis: involvement of glia in enteric and central nervous system”. Your comments and suggestions helped to greatly improve the manuscript.
Reviewer 2 Report
<br class="Apple-interchange-newline">The authors have partially improved the manuscript with the suggestions made above. But they have not done any of the suggested experiments that I consider essential.
English is ok
Author Response
Dear Editor and Reviewers,
We would like to thank you for the second revision of our manuscript titled “Anti-hyperalgesic efficacy of acetyl l-carnitine (ALCAR) against visceral pain induced by colitis: involvement of glia in enteric and central nervous system”. We have revised and improved the manuscript accordingly. All authors have approved the response letter and the revised version of the manuscript.
Reviewer 2
The authors have partially improved the manuscript with the suggestions made above. But they have not done any of the suggested experiments that I consider essential.
Response. We are grateful to the Reviewer for the suggestions which helped a lot in improving the manuscript. We think the experiments suggested by the Reviewer might be part of a second work dedicated to the study of the mechanisms behind ALCAR effects, for which more experiments than those suggested are needed. Indeed, conventional imaging methods for assessing microglial morphology are still inconsistent (doi: 10.1038/s41598-022-23091-2) and they do not provide sufficient information about microglia phenotype which has been demonstrated to be heterogenous, despite the similarities in the morphology (doi: 10.3390/cells11152383; doi: 10.1002/iid3.362; doi: 10.1016/j.celrep.2020.01.010). Only the gene expression pattern of the disease-associated clusters might properly describe the functional changes occurring in the activated microglial cells. This analysis would require dedicated experimental setups which need to be validated in vivo to demonstrate the involvement in pain modulation. Yet, ALCAR is involved in several other signalling pathways, regulating the expression of several proteins as well as the activity of receptors involved in pain physiology. Therefore, the repeated administration with ALCAR in the context of visceral pain might be the results of several concomitant changes. The results we can obtain by investigating a single factor could led to misleading conclusions. On the other hand, changes in the expression of receptors/proteins (already demonstrated in literature) will not definitely demonstrate or exclude any mechanisms beyond the effect of ALCAR. Demonstrating the mechanisms requires dedicated functional studies which, despite they are beyond the aim of the present work, might feed into future researches.
Minor editing of English language required.
Response. We revised the English language according to Reviewer suggestion.
The introduction, the cited references, the research design, the methods description, the presentation of the results and the conclusions can be improved.
Response. We performed a second revision of the manuscript in order to improve it according to Reviewer comments.
